# The Effects of Selenium on Wheat Fusarium Head Blight and DON Accumulation Were Selenium Compound-Dependent

**DOI:** 10.3390/toxins12090573

**Published:** 2020-09-06

**Authors:** Xueyun Mao, Chen Hua, Liang Yang, Yuhui Zhang, Zhengxi Sun, Lei Li, Tao Li

**Affiliations:** Key Laboratory of Plant Functional Genomics of the Ministry of Education, Jiangsu Key Laboratory of Crop Genomics and Molecular Breeding, Collaborative Innovation of Modern Crops and Food Crops in Jiangsu, Jiangsu Key Laboratory of Crop Genetics and Physiology, College of Agriculture, Yangzhou University, Yangzhou 225009, China; xueyun-mao@foxmail.com (X.M.); chenhua0729@foxmail.com (C.H.); yangliang0702@foxmail.com (L.Y.); yuhuizhang123@foxmail.com (Y.Z.); 007179@yzu.edu.cn (Z.S.); livelei@163.com (L.L.)

**Keywords:** wheat, *Fusarium graminearum*, DON, Selenium compounds

## Abstract

Fusarium head blight (FHB) caused by *Fusarium graminearum* not only results in severe yield losses, but also contaminates wheat grains with deoxynivalenol (DON) toxins. Prevention and control of FHB and DON contamination rely mainly on resistant varieties and fungicides. Selenium (Se) is an essential element for humans and animals, and also a beneficial element for plants. In this work, four Se compounds, i.e., sodium selenite (Na_2_SeO_3_), sodium selenate (Na_2_SeO_4_), selenomethionine (SeMet) and selenocysteine (SeCys_2_), were supplemented in a trichothecene biosynthesis induction (TBI) solid medium at different dosages in in vitro experiments. The four Se compounds at the dosage of 20 mg∙L^−1^ were sprayed onto wheat spikes immediately after inoculation at anthesis. All four of the Se compounds significantly inhibited the mycelial growth and DON production in the in vitro experiment; however, in planta, their effects on FHB severity and toxin accumulation in grains were compound-dependent. SeMet consistently negatively regulated fungal growth and DON accumulation both in vitro and in planta, which could be a novel and proconsumer strategy for reducing the detriment of wheat FHB disease and DON accumulation.

## 1. Introduction

Fusarium head blight (FHB) caused by *Fusarium graminearum* Schwabe *(F. graminearum)* is a devastating disease of wheat worldwide [1]. It not only results in severe yield losses, but also contaminates wheat grains with various mycotoxins, such as deoxynivalenol (DON) and its derivatives (3-Acetyl-DON, 15-Acetyl-DON and DON-3-glucoside) [2], causing serious harm to humans and animals. Prevention and control of wheat FHB and toxin contamination have been attracting extensive attention from the scientific community, food and feed processing industries, and end consumers. In addition to breeding resistant cultivars, chemical control is the most commonly used method of prevention and control of the disease. However, conventional chemical control may cause fungicide residues and environmental pollution. There have been reports that the pathogen may develop resistance to fungicides that are frequently applied. Although most commercial fungicides were able to inhibit conidial growth and reduce fungal biomass, some induced DON production [3,4]. Therefore, developing efficient alternative control measures that are environmentally-friendly and healthy against the disease and toxin accumulation has been prioritized.

Selenium (Se) is an essential trace element for humans and animals. It plays an important role in antioxidation and human immunity, and deficiency of Se is associated with many human diseases [5,6,7]. Se is also a beneficial element for plants; the application of Se fertilizer promotes plant growth and improves tolerance to oxidative stresses [8,9]. Se exists in different compounds in plants, which can convert inorganic Se into organic forms dominated by selenomethionine (SeMet) [10]. Organic forms have less toxicity to humans and animals than inorganic forms.

Inorganic Se forms, such as sodium selenite (Na_2_SeO_3_) or sodium selenate (Na_2_SeO_4_), have inhibitory effects on fungi or synergistic effects with fungicides [11,12,13]. Both inorganic and organic Se forms were recently reported to inhibit *F. graminearum* growth in vitro [14]. Chemically synthesized Se-glucose was able to inhibit the generation of DON toxin [15]. However, the effects of different Se forms on the control of FHB and mycotoxin production in planta have not been reported.

In this study, we validated the effects of different Se-compounds on both mycelial growth and DON accumulation in vitro, and also deciphered their effects on FHB severity and DON concentrations in infected grains in planta, aiming to explore a strategy which is not only effective in the prevention and control wheat FHB and toxin accumulation, but is also beneficial to the health of plants, humans, and animals.

## 2. Results

### 2.1. Se Inhibits the Growth of F. graminearum and DON Generation In Vitro

Presupplementation of different Se compounds in TBI media delayed the growth of the fungus, and the diameter of the colonies became smaller when compared with the control. Under the same Se dosage, SeMet had the best inhibitory effect, followed by Na_2_SeO_4_ and SeCys_2_. Na_2_SeO_3_ had a weaker effect than the other three forms. The diameter of the colony decreased with increased concentrations of Se compounds, with the exception of SeCys_2_. (Table 1, Appendix A). After 96 h of culture (Table 1), DON was detected only in the medium of the control, but was not in any of the media containing Se supplementation.

### 2.2. In Planta Effects of Se on FHB Severity and Toxin Concentration Were Compound-Dependent

When compared with the control, SeMet treatment lowered both disease severity (evaluated by the proportions of scabbed spikelets, PSS) and DON content in grains; in contrast, Na_2_SeO_3_ treatment increased PSS and DON content in the grains. Na_2_SeO_4_ treatment decreased DON content in the grains but had a similar effect on disease severity to the control. SeCys_2_ treatment reduced disease severity but had an equivalent DON concentration to the control (Table 2, Appendix A).

Se was not detected in the grains of the control, but was detected in grains after foliar application of different compounds, with contents varying depending on the Se compound, indicating that Se was deficient in the soil. Se contents in the grains with inorganic Se treatment were higher than those with organic Se treatment. The lowest Se content was detected with SeCys_2_ treatment (Table 2). There were no significant correlations among PSS, DON content and Se content (*p* > 0.05) (Appendix A).

## 3. Discussion

In this study, the effects of four Se compounds on the growth of *F. graminearum*, FHB severity and DON content in grains were compared both in vitro and in planta. All four of the Se compounds showed significant inhibitory effects on both mycelial growth and DON production in vitro, whereas in planta, they showed compound-dependent effects on disease severity as well as DON content in grains. Interestingly, SeMet treatment consistently showed strongly inhibitory effects on fungal growth and DON accumulation both in vitro and in planta. In contrast, Na_2_SeO_3_ application increased disease severity and DON accumulation in planta, inconsistent with its inhibitory roles in pathogen growth in vitro. Na_2_SeO_4_ treatment showed much stronger inhibitory effects than Na_2_SeO_3_ on mycelial growth in vitro and DON generation in planta. In the in vitro experiment, the inhibitory effects of the four Se compounds on mycelial growth were consistent with the work recently reported [14]. Since organic forms were more effective in inhibiting pathogen growth and DON accumulation, it was speculated that inorganic Se might need to be converted into organic Se in plants to play an inhibitory role. Na_2_SeO_4_ can be assimilated more efficiently by plants than Na_2_SeO_3_ [16], and the discrepancy of the two inorganic forms in the inhibition rate might be due to differences in transport and assimilation efficiency [17].

Wheat infected by *F. graminearum* induced the production of free radicals, which, in turn, induced pathogens to produce toxins [18]. Se is an essential and core constituent of glutathione peroxidase, which plays a key role in antioxidation and can scavenge excessive free radicals [19]. It was inferred that SeMet may be superior to the other three Se forms in scavenging free radicals, resulting in reduced induction of toxins, which might partially explain the different regulatory effects of the four Se compounds on FHB severity. In the current study, the effects of Se on mycelial growth in vitro, as well as on FHB severity and DON production in planta, were obviously Se compound-dependent. Unexpectedly, all four Se compounds suppressed DON production in the in vitro experiment, suggesting that different mechanisms might be involved between in vitro and in planta experiments, and more work is needed to understand the mechanisms of the different Se compounds involved in FHB severity and DON production.

Wheat accounts for 20% of the world’s calorie consumption, and it is also a cereal crop that has a high potential for the accumulation of Se [20]. Se in wheat grains exists mainly in the form of SeMet [21]. Although SeMet is much safer than inorganic Se due to its lower toxicity compared to inorganic Se, its high cost makes it impractical as a large-scale fungicide for the control of FHB. In Se-deficient soil, the use of relatively cost-effective inorganic Se (Na_2_SeO_4_), which can be transformed into SeMet in wheat, is recommended. Since most wheat production regions in China and worldwide are Se-deficient in soil, Se-fortification in these regions would contribute to the prevention and control of FHB epidemics and DON contamination. It is recommended that Se be applied prior to the flowering period, when wheat is the most sensitive to *F. graminearum* infection. This guarantees the transformation of inorganic Se into the SeMet that is to be accumulated in wheat spikes and grains, which combines FHB and DON control with Se enrichment to produce safe and healthy food.

## 4. Materials and Methods

### 4.1. Effect of Se on the Growth of F. graminearum and DON Accumulation In Vitro

*F. graminearum* isolate PH-1, kindly provided by Dr. Bing Li from Yangzhou University, China, was used in this study. Four Se compounds, comprising two inorganic forms (Na_2_SeO_3_ and Na_2_SeO_4_) and two organic forms (SeMet and SeCys_2_) (all were purchased from TCI Development Co., Ltd., Shanghai, China), were used in the in vitro and in planta experiments.

Four dosages of Se, i.e., 0, 20, 40 and 80 mg.L^−^^1^ (concentration of pure Se), for each Se compound were chosen for the in vitro experiment. *F. graminearum* isolate PH-1 was firstly activated on potato dextrose agar (PDA) medium for 72 h and then cleaved into round fungal blocks of 6 mm in diameter with a puncher. One block of fungal inoculum was placed onto the center of trichothecene biosynthesis induction (TBI) [22] solid medium with prior supplementation of the four Se-compounds, with three replicates per treatment. The medium for each was then incubated at 25 °C in an incubator. The diameter of the mycelial colony was measured by the cross method using a vernier caliper. The inhibition rate (IR) of mycelial growth was calculated by IR = (Control-Treatment)/Control × 100%). After 96 h of culture, all media were finely crushed with a 60 m-nylon mesh to determine the DON content in the medium.

### 4.2. Effects of Se Forms on FHB and Toxin in Grain in Planta

The wheat line NIL-7 (selected from the derivatives of the cross of Sumai3 and NP516) with moderate resistance to FHB was planted in the experimental field of Yangzhou University, China. Each Se compound treatment had two replicates and 30 plants (with 2 rows and 15 plants per row) per replicate. The plot size of each replicate was 0.9 m^2^ (1.5 m × 0.6 m). The plant density was about 3 cm^2^ per plant.

Preparation of conidial spore suspension was carried out following the protocol described by Li et al. [23]. Inoculation was carried out using the single floret inoculation method at anthesis (spore concentration is 10^5^/mL, 20 µL per floret), and Na_2_SeO_3_, Na_2_SeO_4_, SeMet, and SeCys_2_ (20 mg∙L^−1^ pure Se, 30 mL per row) were evenly sprayed immediately after inoculation using Pistol Grip Sprayer (Amway Home, Guangzhou, China). The inoculated spikes were then covered with a plastic sandwich bag to sustain humidity; the bag were removed 72 h after inoculation. The proportions of scabbed spikelets (PSS, calculated by infected spikelets/total spikelets) was determined 25 days after inoculation. The contents of DON and Se in grains were determined immediately after harvest.

### 4.3. Extraction and Quantification of DON

The DON content in the samples (1 g) was extracted with 4 mL of acetonitrile-formic acid-water. Extracts were filtered through 0.45 µm and 0.22 µm needle-type organic phase filters. All extracts were stored at −20 °C for subsequent experiments. Toxin concentration was determined using a liquid chromatography triple-quadrupole mass spectrometry instrument (TSQ Vantage, Thermo Fisher Scientific, Waltham, MA, US). The limit of detection was 1 µg.kg^−1^. The parameters of liquid chromatography and mass spectrometry are detailed in Appendix A (Technical note).

### 4.4. Determination of Se Content in Grains

The total Se in grains was determined following the protocol described in [24], using concentrated HNO_3_ and H_2_O_2_ digestion followed by inductively coupled plasma-optical emission spectroscopy (ICP-OES) (X series2, THERMO SCIENTIFIC) analysis.

### 4.5. Data Analysis

Correlation and variance analysis were conducted using SPSS statistics 23 (SPSS Inc., Chicago, IL, USA).

## Figures and Tables

**Table 1 toxins-12-00573-t001:** Inhibitory effects of different Se compounds in different dosages on *F. graminearum* growth and DON accumulation.

Concentration (mg∙L^−1^)	Treatment	Inhibition Rate (%)	DON Content (µg∙kg^−1^)
0	CK	-	156.49
20	Na_2_SeO_3_	29.22 ± 1.81	NF
	Na_2_SeO_4_	66.53 ± 1.22	NF
	SeMet	82.02 ± 1.88	NF
	SeCys_2_	62.72 ± 0.88	NF
40	Na_2_SeO_3_	58.52 ± 2.27	NF
	Na_2_SeO_4_	84.77 ± 1.39	NF
	SeMet	88.64 ± 0.40	NF
	SeCys_2_	60.77 ± 0.26	NF
80	Na_2_SeO_3_	67.90 ± 1.32	NF
	Na_2_SeO_4_	87.09 ± 1.05	NF
	SeMet	92.94 ± 0.00	NF
	SeCys_2_	62.18 ± 0.16	NF

CK, control group; NF, not found.

**Table 2 toxins-12-00573-t002:** FHB severity, DON content and Se content in grains treated with the different Se-compounds.

Treatment	PSS	DON Content (µg∙kg^−1^)	Se Content (mg∙kg^−1^)
CK	0.418 ± 0.036 b	320.45 ± 12.23 c	0 ± 0.000 a
Na_2_SeO_3_	0.552 ± 0.004 c	686.30 ± 28.29 d	2.656 ± 0.113 c
Na_2_SeO_4_	0.463 ± 0.027 b	223.60 ± 28.28 b	2.815 ± 0.147 c
SeMet	0.276 ± 0.002 a	144.28 ± 2.45 a	2.471 ± 0.313 c
SeCys_2_	0.323 ± 0.009 a	339.94 ± 26.93 c	0.878 ± 0.007 b

CK, control group; a multiple comparison post hoc test was implemented, and different letters after the value indicate there were significant differences between the values at *p* = 0.05, and same letter indicates they were statistically same. The proportion of scabbed spikelets (PSS) was evaluated at 25 days after inoculation, and DON content was determined immediately after harvest.

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
