# Peer review of "The Effects of Selenium on Wheat Fusarium Head Blight and DON Accumulation Were Selenium Compound-Dependent"

_toxins, 2020, doi:10.3390/toxins12090573_

Round 1
Reviewer 1 Report
The presented manuscript is an interesting work due to it has been investigated four Se compounds including sodium selenite (Na2SeO3), sodium selenate (Na2SeO4), 10 selenomethionine (SeMet) and selenocysteine (SeCys2) at different dosages supplemented in trichothecene biosynthesis induction (TBI) solid medium in vitro because which could be a strategy for reducing the detriment of wheat FHB disease and DON accumulation.
However minor revisions are required. According that and after minor revision I recommend the manuscript to be published. Below it is detailed few points to be consider:
Please explain with detail the time of incubation and the time of determination of DON during the exposure concentration studied. Then explain the results observed during the exposure times. In Table 2 is not indicated at what time it has been evaluated, if after 25 days or at 96h.
In line 114: please replace “in vitro” by “in vitro” (cursive format)
In lines 120: It is detailed three replicates per treatment then please in table 1 include the standard deviation of the results.
In line 141-143: It is not indicated the determination procedure in this section and neither in the article. Please include conditions of chromatography and mass spectrometry parameters including precursor and product ion.
Author Response
Reviewer 1
Comments and Suggestions for Authors
The presented manuscript is an interesting work due to it has been investigated four Se compounds including sodium selenite (Na2SeO3), sodium selenate (Na2SeO4), 10 selenomethionine (SeMet) and selenocysteine (SeCys2) at different dosages supplemented in trichothecene biosynthesis induction (TBI) solid medium in vitro because which could be a strategy for reducing the detriment of wheat FHB disease and DON accumulation.
However minor revisions are required. According that and after minor revision I recommend the manuscript to be published. Below it is detailed few points to be consider:
- Please explain with detail the time of incubation and the time of determination of DON during the exposure concentration studied. Then explain the results observed during the exposure times. In Table 2 is not indicated at what time it has been evaluated, if after 25 days or at 96h.
REPLY:Thanks! In vitro experiment, the content of DON was determined in 96 h after incubation. In planta experiment, the proportion of scabbed spikelets was evaluated in 25 days after inoculation. The contents of DON in grains were determined immediately after harvest. These details were added in Table 2.
- In line 114: please replace “in vitro” by “in vitro” (cursive format)
REPLY:Thanks! Changed as suggested.
- In lines 120: It is detailed three replicates per treatment then please in table 1 include the standard deviation of the results.
REPLY:Thanks! The standard deviation of the results has been added to Table 1.
- In line 141-143: It is not indicated the determination procedure in this section and neither in the article. Please include conditions of chromatography and mass spectrometry parameters including precursor and product ion.
REPLY:Thanks! The detailed conditions of chromatography and mass spectrometry parameters were described in Supplementary file 1.
Reviewer 2 Report
A previous work carried out under the same experimental conditions as described in the in vitro experiment in this manuscript was recently published and should be cited.
Results
Line 60: What was the limit of detection of Deocynivalenol (DON)? This might also explain the lack of DON detection in the plates.
Line 73: Where is the correlation analysis?
Discussion
This section should be improved and limitations of the study highlighted.
In vitro experiment: A previous study detected DON level of 600ppb in their control plate of Fg only without Se amendment. However, in this manuscript, the concentration of DON reported in the control (Fg only without Se amendment) was 156.49 ug kg-1, Both experiments were carried under the same experimental conditions and utilised same resources including Fg isolate, Fg activation and growth medium, incubation period, incubation temperature/condition, size of inoculum plug etc. Such significant variation in the results should be discussed. In addition, the inhibition rate were similatr for example SeMet in both studies despite the difference in Se concentration. This should be also be addressed.
Line 92-107: Reference to the findings in this manuscript should also be included in this section of discussion.
Line 104-106: What was the basis of your recommendation to spray Se before flowering when your findings were based on Se application at anthesis after inoculation with Fg (Line 131-133)?.
Materials and Methods
Line 115: What was the rationale for selecting Se concentration of 20, 40 and 80 mg/L?
Line 121: how long after inoculation was the diameter of the mycelial colony measured and and how frequently was it measured?
Line126-128: The experimental field was described as 'two rows and 15 plants per row'. Is this overall size of the experimental field? How many replicates per treatment? What was the plot size per treament?, what was the plant density per treatment?
Line 130: The reference cited is incorrect. In fact, reference [23] cited a review paper. You should describe the protocol briefly or include the correct reference.
Line 132: Please indicate briefly why plants were treated with 20 mg L-1 Se. Was this based on the output from the in vitro experiment?
Line 135: How many wheat ears were assessed for disease and processed for DON detection per treatment. ?
Line 139: What was the limits of DON detection in your assay?.This is relevant particularly in terms of the in vitro experiment. Detection of DON may be constrained by your assay detection limit. I would expect that you should detect some levels of DON with the selenium treatment based on the wide margin in the rate of inhibition of Fg growth presented in table 1.
Line 139: Reference cited is incorrect, it is a review paper with no reference to DON extraction protocol. Please cite a paper that actually describes the DON extraction method you adopted.
There is no information on statistical analyses used in this study!
Author Response
Comments and Suggestions for Authors
A previous work carried out under the same experimental conditions as described in the in vitro experiment in this manuscript was recently published and should be cited.
REPLY:Thanks! We have cited the designated relevant work both in the Introduction and Discussion sections.
Results
- Line 60: What was the limit of detection of Deocynivalenol (DON)? This might also explain the lack of DON detection in the plates.
REPLY:Thanks! The limit of determination of DON is 1 µg.kg-1 (1 ppb).
- Line 73: Where is the correlation analysis?
REPLY:Thanks! Since the correlations were not significant between the components (P>0.05), we did not include the detailed information in the previous version. Here the correlation analysis was attached in Supplementary Table S1.
- In vitro experiment: A previous study detected DON level of 600 ppb in their control plate of Fg only without Se amendment. However, in this manuscript, the concentration of DON reported in the control (Fg only without Se amendment) was 156.49 ug kg-1, Both experiments were carried under the same experimental conditions and utilised same resources including Fg isolate, Fg activation and growth medium, incubation period, incubation temperature/condition, size of inoculum plug etc. Such significant variation in the results should be discussed. In addition, the inhibition rate were similar for example SeMet in both studies despite the difference in Se concentration. This should be also be addressed.
REPLY:We really appreciate the suggestions. We did not cite the work published in the conference proceedings mainly because peer-reviewed journals seldomly cite the work without peer-reviewing process. In the previous study, the toxin content was determined via enzyme-linked immunosorbent assay kit. The assay kit was somewhat sensitive to the fluctuations in room temperature and also to operating skills, which sometimes resulted in large variations. In this study, we used LC-MS/MS to determine the content of DON in grains, which significantly improved robustness and reliability of the assay.
- Line 92-107: Reference to the findings in this manuscript should also be included in this section of discussion.
REPLY:Thanks! We have improved the discussion as suggested.
- Line 104-106: What was the basis of your recommendation to spray Se before flowering when your findings were based on Se application at anthesis after inoculation with Fg (Line 131-133)?
REPLY:Very good question! SeMet had the best inhibitory effect both in vitro and in planta experiments when compared with the other three Se compounds. Commercial SeMet is very expensive. Wheat can convert inorganic Se into organic forms, predominant by SeMet, therefore, we recommended spraying inorganic Se before flowering stage, when is the most sensitive to Fusarium head blight disease. However, more experiments are needed to evaluate the effects of endogenous and exogenous SeMet on disease control.
Materials and Methods
- Line 115: What was the rationale for selecting Se concentration of 20, 40 and 80 mg/L?
REPLY:A nice question! Based on our previous experiments, 20 mg/L of Se compounds had good inhibitory effects. However, we were interested in knowing whether increase in the dosage had better or completely inhibitory effects on pathogen control, we doubled and quadrupled the dosage. Considering the cost, we used 20 mg/L of Se compounds in planta experiment.
- Line 121: how long after inoculation was the diameter of the mycelial colony measured and how frequently was it measured?
REPLY:Thanks! The first measurement was performed 24 h after inoculation, and then measured every 24 h.
- Line 126-128: The experimental field was described as 'two rows and 15 plants per row'. Is this overall size of the experimental field? How many replicates per treatment? What was the plot size per treatment? What was the plant density per treatment?
REPLY:Thanks! each treatment had two replicates and 30 plants per replicate. The plot size of each replicate was 0.9 m2 (1.5 m * 0.6 m). The plant density was 3 cm2/per plant.
- Line 130: The reference cited is incorrect. In fact, reference [23] cited a review paper. You should describe the protocol briefly or include the correct reference.
REPLY:Sorry for incorrect citation. We added detailed protocol (Supplementary file 1) developed in our lab recently.
- Line 132: Please indicate briefly why plants were treated with 20 mg L-1 Se. Was this based on the output from the in vitro experiment?
REPLY:A nice question! In vitro results showed that 20 mg L-1 had a good inhibitory effect. Additionally, considering the cost, we used 20 mg/L of Se compounds in planta experiment.
- Line 135: How many wheat ears were assessed for disease and processed for DON detection per treatment?
REPLY:Thirty wheat ears per replicate were assessed for disease severity and processed for DON detection per treatment.
- Line 139: What was the limits of DON detection in your assay? This is relevant particularly in terms of the in vitro experiment. Detection of DON may be constrained by your assay detection limit. I would expect that you should detect some levels of DON with the selenium treatment based on the wide margin in the rate of inhibition of Fg growth presented in table 1.
REPLY:Thanks! The limit of determination of DON is 1 µg/kg. We had similar expectation as the reviewer’s, however, the results were unexpected.
- Line 139: Reference cited is incorrect, it is a review paper with no reference to DON extraction protocol. Please cite a paper that actually describes the DON extraction method you adopted.
REPLY:Thanks! We added a detailed protocol (Supplementary file 1) developed in our lab recently.
- There is no information on statistical analyses used in this study!
REPLY:Thanks! the statistical analyses was added in the method section.
Round 2
Reviewer 2 Report
The authors have improved their paper in response to the comments made and have addressed major concerns. However, i have a few additional minor comments for the authors to consider.
Line 151: remove statement in parenthesis
Line 152: remove parenthesis and its content
Line 152/153: Please remove ‘/’
Line 165 I suggest rephrase the sentence to ‘DON content in samples (?g) were extracted with ? ml of acetonitrile-formic acid-water.
Line 177: indicate the post hoc test used.
Author Response
Reviewer 2
The authors have improved their paper in response to the comments made and have addressed major concerns. However, i have a few additional minor comments for the authors to consider.
Line 151: remove statement in parenthesis
REPLY: Thanks! “(49.5: 1: 49.5)” was removed as suggested
Line 152: remove parenthesis and its content
REPLY: Thanks! “(LC-MS/MS)” was removed as suggested
Line 152/153: Please remove ‘/’
REPLY: Thanks! ‘/’ was removed as suggested
Line 165 I suggest rephrase the sentence to ‘DON content in samples (?g) were extracted with ? ml of acetonitrile-formic acid-water.
REPLY: Thanks! As suggested, we rephrased the sentence to ‘DON content in samples (1 g) were extracted with 4 mL of acetonitrile-formic acid-water.
Line 177: indicate the post hoc test used.
REPLY: Thanks! we put “A multiple comprison post hoc test was implemented” in the legend of Table 2